# Allergy to Cow’s Milk Proteins and Other Allergens—An Unrecognized Co-Factor of Idiopathic Nephrotic Syndrome in Children or a Factor Interfering with the Treatment of This Disease? A Case Report

**DOI:** 10.3390/reports6020025

**Published:** 2023-05-25

**Authors:** Maciej Kaczmarski

**Affiliations:** Department of Paediatrics, Gastroenterology and Allergology, Medical University of Bialystok, 15-089 Bialystok, Poland; maciej.kaczmarski@umb.edu.pl or maciej.g.kaczmarski@gmail.com

**Keywords:** idiopathic nephrotic syndrome (INS), atopy, immunoglobulin E (IgE), interleukin 13 (IL-13), atopic/allergic diseases, cow’s milk allergy, cross-allergy, allergens, elimination diet, anti-allergic drugs, children

## Abstract

Idiopathic nephrotic syndrome (INS) is one of the chronic kidney diseases that occurs in childhood. Starting from a few case reports in the 1950s–1970s and up to the present, the relationship between idiopathic nephrotic syndrome (INS) and the occurrence of atopic disorders in these patients has been discussed in many medical publications. These publications show that in certain patients, mainly children and adolescents, but also in adults with INS, various clinical symptoms and laboratory indicators of an atopic allergic process may be present. This process has been shown to involve Th2 lymphocytes, to have an excessive production of interleukins (IL-4, IL-5, IL-13), and to have an increased serum level of immunoglobulin E (IgE). This leads to the development of a systemic allergic inflammatory process, of which the kidneys can also become the effector organ. The coexistence of an allergic process which may adversely affect the course of nephrotic syndrome may be confirmed by the increased serum IgE level and the hypersensitivity reaction of the patient’s body to various environmental allergens (through the presence of allergen-specific IgE /asIgE/ antibodies to food, pollen, mould, dust, or other allergens in the blood serum). High concentrations of IL-13 and other plasma mediators of this inflammation (e.g., histamine, bradykinin) structurally and functionally damage the renal filtration barrier, and in particular, the function of podocytes in the glomeruli. Podocyte dysfunction disturbs the physiological process of plasma filtration in the glomeruli, leading to excessive protein loss in the urine. These disorders initiate the development of idiopathic nephrotic syndrome in these patients. This publication presents the coexistence of an allergic process caused by allergy to cow’s milk proteins and hypersensitivity to other allergens in a child with idiopathic nephrotic syndrome. This publication also assesses whether treatment of the allergic process coexisting with INS with an elimination diet (milk-free, hypoallergenic) and anti-allergic drugs affected the course and treatment of INS in this child.

## 1. Introduction

The term idiopathic nephrotic syndrome (INS) is often used to describe a heterogeneous group of glomerulopathies that occur predominantly in children [1]. The disease is defined by the presence of massive proteinuria, hypoalbuminemia, hyperlipidaemia, and oedema [2,3].

In most children with INS, histologically confirmed changes in the glomeruli are minimal glomerular lesions; therefore, INS may also be synonymously viewed as minimal change disease (MCD) [3,4]. Depending on the response to corticosteroids, INS is currently classified as steroid-sensitive idiopathic nephrotic syndrome (SSINS) or steroid-resistant idiopathic nephrotic syndrome (SRINS) [3,4]. MCD is the most common type of INS that is responsive to steroid treatment [3,5]. 

In many publications, both earlier (from the 1950s–1980s) and more recent, authors have discussed the causal relationship between INS and the occurrence of atopic disorders in patients. These publications show that in certain patients with INS, mainly children and adolescents, but also adults, the presence of various clinical symptoms and laboratory indicators typical of an ongoing atopic allergic process may be present. However, the nature of these relationships remains unclear [2,3,4,5,6].

They refer thematically to case reports from the 1950s, which described the occurrence of allergic complaints in children allergic to poison ivy and with nephrotic syndrome (Rytand), in children with allergic skin lesions and INS (Fanconi), and in adults with hypersensitivity to airborne allergens and INS (Hardwicke) [6,7].

The authors of articles published in later years also observed in certain patients an association between INS and hypersensitivity to various environmental allergens (food allergens, inhalant allergens) with reactions after vaccinations and after insect stings or hymenoptera stings [8,9,10,11,12,13,14]. 

A significant number of these patients also suffered from other diseases pathogenetically associated with an atopic background: asthma, atopic dermatitis, and allergic rhinitis/conjunctivitis. Most of these allergic diseases appeared within 2–6 months after INS developed [2,4,5,6,8,15,16].

Allergic symptoms occurring in the course of these diseases arise as a result of increased activity of Th2 lymphocytes and the production of cytokines IL-4, IL-5, and IL-13, involved in the synthesis of immunoglobulin E (IgE) [6,17,18,19].

The results of published studies also confirm the frequent occurrence of elevated levels of IgE in the serum of INS patients (50–70%), compared to age-matched allergic patients and control subjects [2,18].

High levels of serum IgE found during the onset of the disease are associated with a worse course of INS, more frequent recurrences, and less effective steroid therapy.

A higher concentration of serum IgE has also been found in patients with INS not responding to steroids (SRINS) compared to steroid-sensitive patients (SSINS), both pre- and post-treatment [2,4,18,19].

The opinion that INS in some patients is a clinical manifestation of an allergy induced by specific allergens is still the subject of scientific discussion [2,4,6].

The essence of INS, regardless of the cause, is always anatomical and functional damage to podocytes. Podocytes are highly specialized epithelial cells of the visceral glomerular capsule. They play an essential role in selective plasma filtration and the formation of primary urine [1,3,20].

Two hypotheses are currently being discussed to explain the mechanism of damage to these cells which leads to the development of nephrotic syndrome [3]. One theory suggests that proteinuria is induced by a primary glomerular defect caused by a mutation in the gene encoding the podocyte structure, or in the glomerular basement membrane proteins, thereby causing podocyte effacement [2,3,21]. Podocyte dysfunction leads to massive loss of protein in the urine (proteinuria) and the clinical manifestation of INS in the form of oedema [2,3].

The second, an immunological hypothesis of the pathogenesis of INS refers to the publication of Shalhoub from 1974, who was the first to suggest T-cell dysfunction in patients with INS, resulting in an increased level of permeability factor in plasma, which affects the shape and function of podocytes [1,3,22,23,24,25].

Recent studies have indicated the pathogenetic role of interleukin 13 (IL-13), the main stimulator of IgE synthesis in the human body, which leads to the development of IgE-dependent allergic inflammation (inflammation type 1) [26,27].

In the allergic process taking place in the human body, T lymphocytes are activated and differentiated into effector cells (T effector). Activated Th2 effector cells release different cytokines, among which IL-13 is included [4,19,24,25].

The high level of serum IgE found in some INS patients is a manifestation of excessive production of IL-13, which also leads to the expression of the costimulatory particle CD80 by podocytes [2,4]. At the same time, these patients have impaired function of regulatory lymphocytes (Tregs). The dysfunction of these cells lies in their inability to produce appropriate cytokines (among others, CTLA-4 and IL-10) in the patient’s body, and in the removal of the stimulating effect of IL-13 for the permanent expression of CD80 by podocytes [1,4,24,28,29].

The biological effect of IL-13, represented by ILC2 lymphoid cells, has also been confirmed in non-IgE-mediated allergy in conjunction with innate immunity [28,30,31]. ILC2 cells produce various cytokines when stimulated by allergens. These include IL-5 and IL-13, which lead to the development of type 2 allergic inflammation in the patient’s body. In this type of allergic inflammation, the destructive role of the subpopulation of ILC2 cells (producing cytokines IL-25, IL-33) results in damage to the epithelial cells of the respiratory tract, digestive tract, and skin keratinocytes, and to the endothelial cells of various effector organs (e.g., kidneys) [28,30,31]. Immunoglobulin E is synthesized in these patients at concentrations below the threshold of detection of standard allergy tests. Therefore, the results of immunoglobulin E titres (tIgE) and allergen-specific antibodies are usually negative [30,31].

Constant expression of CD80 particles disturbs both the anatomical structure of podocytes (these changes are confirmed by electron and light microscopy) and their function [1,4]. The results of disturbed filtration of plasma proteins in the renal glomeruli are the clinical and laboratory symptoms of INS [2,3,4,5,6].

Taking into account the cited studies, it seems that the role of allergic hypersensitivity as a potential factor involved in the development of idiopathic nephrotic syndrome is debatable in terms of etiopathogenetic and therapeutic aspects.

## 2. Case Presentation

The subject of this case study is a girl aged 6 years and 9 months, born from a twin pregnancy (twin I) and by caesarean section. The condition of the child after birth was good, with a birth weight of 3100 g, Apgar scale 10 points. Occurrence of allergic diseases in the family was confirmed; both parents had symptoms of an inhalant allergy, and the girl’s twin brother was treated with a milk-free diet until the age of 3 years due to a cow’s milk protein allergy. The girl was breastfed for 2 months and then fed a modified cow’s milk formula.

According to the mother, the girl had recurrent upper respiratory tract infections (every 1.5 months), consisting predominantly of a cough and persistent runny nose, from the age of 9 months until the onset of nephrotic syndrome at the age of 2.5 years. These symptoms were not always accompanied by a fever.

Due to an infection (fever, runny nose, cough, and bronchial obstruction with wheezing), the child was admitted to the allergology department of the children’s hospital with suspected asthma in January 2019.

During hospitalization, the child also developed swelling of the face, lower limbs and abdominal pain in addition to the symptoms of a respiratory tract infection. Laboratory tests showed hypoproteinaemia (total protein—50.0 g/L, albumin—22.0 g/L), hyperlipidaemia (cholesterol—9.61 mmol/L, triglycerides—2.80 mmol/L), proteinuria (4.2 g/L), CRP < 5.0 g/L, Vit.D_3_—6.86 ng/mL (extremely low value), and IgE—380.1 IU/mL (normal value < 60.0 IU/mL). 

Nephrotic syndrome was diagnosed, and the child was transferred to the department of nephrology, where intravenous treatment with methylprednisolone (2 mg/kg body weight) was commenced. Due to the worsening of respiratory symptoms (pneumonia) and confirmation of concomitant infection with an influenza A virus, an antiviral drug (Oseltamivir) was added to the treatment. The child was transferred for further treatment to the infectious disease ward with a diagnosis of idiopathic nephrotic syndrome, submicroscopic glomerulopathy, pneumonia, and influenza A. The following treatment was recommended upon discharge from the hospital: Prednisone—30 mg/day, inhalations (bronchodilator), oxymetazoline, and a balanced normocaloric diet.

Despite the use of the recommended INS treatment, the child had subsequent relapses of INS symptoms and further hospitalizations. 

In April 2019, laboratory tests showed a proteinuria of 14.5 g/L. Her INS treatment comprised methylprednisolone (intravenously), followed by oral Prednisone 30.0 mg/day. Her allergic symptoms included a runny nose, watery/teary eyes (increased lacrimation), and ocular pruritis. Her serum IgE-level was 380.1 IU/mL, and an inhalant allergy panel gave a negative results. The prescribed anti-allergic treatment consisted of eye drops, and a balanced normocaloric diet was further recommended.

In May 2019, laboratory results showed a proteinuria of 1.71 g/L. Allergic symptoms included a runny nose and cough (without fever). INS treatment then consisted of Prednisone (15.0 mg/day) + Mycophenolate mofetil [MMF] (2 × 1.8 mL) and a balanced normocaloric diet.

From June to November 2019, the child was hospitalized three times with the diagnosis of steroid-dependent nephrotic syndrome. INS treatment consisted of Prednisone (30 mg/day) + MMF (2 × 1.8 mL), but without any significant improvement. 

During the seventh hospitalization (November 2019), treatment with Cyclosporine was proposed, but the parents declined.

From November 2019 to November 2020, the child experienced seven consecutive episodes of INS exacerbations (proteinuria episodes) and was treated exclusively in the outpatient clinic. Prednisone was used in the treatment, in various doses from 30.0 to 5.0 mg/day (in proteinuria episodes) + MMF (2 × 1.8 mL), and periodically, an anti-allergic drug was instituted.

In April 2020 (birch flowering and pollen season), the girl’s parents observed the occurrence of allergic symptoms of a runny nose, watery eyes and conjunctivitis after a walk outdoors, and concurrently, a proteinuria of 1.4 g/L. The child was given an anti-allergic drug. In September, after eating pancakes with white cheese, a proteinuria of 1.14 g/L was noted along with a runny nose and cough. In October, after eating cheese pizza, laboratory tests showed a proteinuria of 1.0 g/L combined clinically with a runny nose.

In addition, the child’s mother noted episodic proteinuria and allergic rhinitis/conjunctivitis occurring after eating apples, bananas, citrus fruits, carrots, leeks, and celery. Anti-allergic drugs were administered at these times.

In December 2020, the girl’s parents, suspecting a correlation of the allergic process with INS exacerbations, sought an allergy consultation with the author of the publication. The child’s age at this time was 4 years and 8 months.

The basis for the consultation was an analysis of the child’s medical records from 2019–2020, which consisted of hospitalization records, results of allergy tests, and an interview with the parents. The child’s nutrition before and during bouts of INS was also analyzed, as well as the type of allergic symptoms observed in the child. A physical examination of the child was also performed.

All the information collected allowed for a preliminary diagnosis of a suspected persistent IgE-mediated allergy to cow’s milk proteins and hypersensitivity to certain non-dairy foods, as well as the suspicion of a pollen and food allergy (due to cross-allergic reactions after eating certain fruits and vegetables). Allergic rhinitis and conjunctivitis were also included in the diagnoses.

The resulting recommendations were as follows: continuation of specialist INS treatment in a paediatric nephrology centre, Prednisone (25 mg/day) + MMF (2 × 1.8 mL), introduction of a balanced milk-free, and a hypoallergenic elimination diet (elimination of certain fruits, vegetables, nuts, and artificially coloured sweets and drinks). Combined with the anti-allergic treatment were loratadine (1 × 5 mg in the morning), ketotifen (1 × 0.5 mg in the afternoon), anti-allergic drugs for the nose and conjunctiva (as necessary), and supplementation with vitamin D_3_ and calcium.

The parents remained in contact with the consulting physician during the treatment.

A follow-up allergy consultation after 9 months of treatment was performed in September 2021 when the child was 5 years and 2 months old. The results of the control allergy tests were as follows: tIgE—140.0 IU/mL, inhalant allergy panel—negative; food allergy panel: cow’s milk proteins—class 2; hen egg white and yolk—class 2. The positive result of the food allergy panel confirmed the initial diagnosis of an IgE-mediated allergy to cow’s milk proteins and to some non-dairy foods (i.e., hen’s egg proteins), and warranted further elimination of these products from the child’s diet.

The reason for further use of the hypoallergenic diet in the child was the suspicion of non-IgE-mediated allergic reactions to some other non-dairy foods.

The effects of the dietary and pharmacological treatment of the allergic process coexisting with INS resulted in

(a) progressive remission of INS symptoms;

(b) reduction of the Prednisone and MMF doses;

(c) discontinuation of Prednisone and MMF treatment (December 2021).

Two episodes of proteinuria were recorded in 2022 after the child consumed apple/carrot juice, and after the child consumed broth containing beef. The proteinuria resolved after the child returned to the proper elimination diet.

As of January 2023, the child remains in remission from INS symptoms. Dietary treatment (a milk-free and hypoallergenic diet) and pharmacological treatment (loratadine 5 mg/day as necessary) for the prevention of allergic symptoms is being maintained in the patient.

The course of INS in the patient before the allergological consultation and after the introduction of the allergist’s recommendations (elimination diet, anti-allergic treatment) is presented in Table 1.

## 3. Discussion

Many medical publications discuss the etiopathogenetic relationship of the simultaneous occurrence of various allergic diseases with idiopathic nephrotic syndrome in certain patients of all age groups including children, adolescents, and adults [2,6,8]. The allergic symptoms found in these patients mainly concern diseases in which immunoglobulin E plays a key pathogenic role (i.e., asthma, bronchial obstruction, allergic rhinitis, conjunctivitis, and others [15,16,26,27,32,33,34].

Similar allergic complaints preceded the occurrence of INS symptoms in the girl with INS who was consulted by the author. The child’s symptoms also persisted during treatment, with episodes of recurrence of symptoms and subsequent hospitalizations. In patients with INS, as described in the cited publications, the occurrence of allergic symptoms is etiopathogenetically associated with sensitization to various environmental, food, and inhalant allergens [6,7,8,9,10,35].

In each patient, the process of sensitization of the body by various allergens is individually differentiated and takes place either with the participation of immunoglobulin E (an IgE-dependent mechanism) or its lack thereof (an IgE-independent mechanism) [26,32,33]. This case of a child with INS who was diagnosed with a coexisting IgE-dependent allergy to cow’s milk and hen’s egg proteins, as described by the author, is in line with the observations of the cited authors, who reported the coexistence of an allergy to cow’s milk proteins in children with INS [9,10,35,36,37,38].

The use of a milk-free diet by Sandberg in children with steroid-sensitive INS and a coexisting cow’s milk allergy resulted in remission from the disease, and allowed for discontinuation of steroid treatment [36]. 

The effectiveness of eliminating cow’s milk proteins and other foods in 12 children with steroid-resistant INS and concomitant IgE-mediated food allergy was also confirmed by Italian authors [37]. The first effect, predicting the remission of INS in these patients, was the rapid resolution of proteinuria during the elimination of the harmful allergen from their diet, and the reappearance of proteinuria when returning to a non-elimination diet [37].

Sieniawska, who treated a group of 17 Polish children with steroid-resistant INS, observed that 6 patients (35.3%) responded to the elimination of cow’s milk proteins from the diet. After a year of using a milk-free diet, their tolerance to cow’s milk proteins was assessed by performing a challenge test with cow’s milk. It was established that two children became tolerant and remained in remission from INS; in another two, remission from the disease was maintained by the continued use of a milk-free diet. The use of a milk-free diet restored sensitivity to steroid therapy in one child with INS [10]. 

The case report presented by the author also concerns a Polish child with steroid-resistant INS. As in the Italian studies and Sieniawska’s studies, and also in the presented case, the first symptom of the effectiveness of the elimination diet in a child with INS was the rapid resolution of proteinuria, which is a sign of disease remission [10,37].

Following the course of the INS disease described by the cited authors, as well as the presented case herein, it can be assumed that the unrecognized allergy to cow’s milk proteins sustained the primary sensitization process of these patients. Persistent sensitization to cow’s milk proteins paved the way for the sensitization of the body by other allergens. The process of successive sensitization may have taken place along with the consumption of various foods, e.g., fruits, vegetables, juices, nuts, artificially colored sweets and drinks, or other non-dairy foods, constituting the basis for the development of cross-allergic reactions (pollen–food) [39,40,41]. This type of reaction may be evidenced by various allergic symptoms found in patients with INS, as described in the cited publications [6,10,16,32]. In the presented case, the possibility of allergic cross-reactions was confirmed by the observations of the child’s mother: the occurrence of proteinuria after a walk next to a dusty birch and after eating an apple (a cross-reaction of birch pollen and apple allergens), and proteinuria after eating broth cooked with beef (a cross-reaction of cow’s milk proteins with beef) [40,41].

The coexisting process of hypersensitivity to cow’s milk proteins and other non-dairy foods should be considered a significant cause interfering with the treatment of INS during the first 2 years of the disease in the consulted patient. It was also a probable cause of the conversion of INS from the steroid-sensitive form (SSINS) to the steroid-resistant form (SRINS). The diagnosis of an allergy to cow’s milk proteins and other foods coexisting with INS and the use of an elimination diet (i.e., the elimination of potentially allergenic foods) in this case became a turning point in the course and treatment of INS.

Within a few months of using a balanced, dairy-free, hypoallergenic diet supported by anti-allergic treatment, long-term remission from INS symptoms was achieved while gradually reducing the dose of Prednisone and Cell-Cept; this ultimately allowed for the discontinuation of these medications.

From December 2020 to January 2023, the girl remained in disease remission with no symptoms, and did not receive medications used to treat INS. She took 5 mg of loratadine for the prevention of allergic symptoms, vitamin D_3_, calcium, and other vitamin supplements. The use of a calorically balanced elimination diet did not cause disturbances in the child’s development, as evidenced by the anthropometric parameters of physical development. As of January 2023, at the age of 6 years and 9 months, her weight was 22 kg (>50th centile) and her height was 119 cm (75th centile).

A subsequent follow-up visit with allergy tests is expected to take place after the plant flowering and pollen season. Planned tests include serum IgE and asIgE against cow’s milk, hen’s egg proteins and other food and inhalant allergens. The obtained results will be an indication for further dietary management.

A review of the cited bibliography unfortunately does not provide any answer to the question of how long an elimination diet should be used in patients with INS and a concomitant allergic process. In the author’s opinion, the moment of discontinuation of dietary treatment in these patients is determined by two factors: the type of sensitizing allergen and the type of sensitizing mechanism of the patient’s body (i.e., IgE-mediated or non- IgE-mediated).

According to Bingol et al., who studied the phenotypes of food allergy in Turkish children, cow’s milk was most often responsible for the development of non-IgE-type allergy [42].

This mechanism of food allergy development has a more favourable prognosis in the process of acquiring immunological food tolerance, because immunoglobulin E does not play a dominant role in the mechanism of the development of allergic symptoms [33].

In the case of IgE-mediated allergy, which was diagnosed in this case report, a high level of serum IgE was an expression of an active allergic process that may delay the acquisition of tolerance to the harmful allergen(s). According to Saarinen’s research, 15.0% of treated children with a previously diagnosed IgE-dependent cow’s milk allergy over the age of 8 years still show symptoms of persistent CMA allergy [43].

The Vanto study showed that infants with a non-IgE-mediated cow’s milk allergy nearly always became tolerant to cow’s milk by 4 years of age [44].

Referring to the biological role of IL-13 and its participation in the synthesis of serum IgE and in the expression of the CD80 molecule by podocytes [17,18,19], in patients with INS and a coexisting allergic process (food allergy, inhalation allergy), remission of kidney disease symptoms can be expected after successful elimination of the harmful allergen from the diet, thus ending the allergic inflammatory process in these patients.

## Figures and Tables

**Table 1 reports-06-00025-t001:** The course of INS in the presented patient before the allergological consultation and after the introduction of the allergist’s recommendations (elimination diet, anti-allergic treatment).

Hospitalizations and Nephrological Consultations	Proteinuria Episodes (Months)	Extent of Proteinuria g/L	Coexisting Allergic Symptoms Other Remarks	INS Nephrological Treatment		Allergy Treatment
				Prednisone (mg)	MMF (mL)	
2019 7 hospitalizations	January /February	4.2–2.2	Onset of INS, influenza A, asthma susp.	January–December starting dose 30.0 per day then various doses	0	Antiallergic treatment used occasionally. Balanced normocaloric diet.
	April	2.74–3.75 14.5	Plant pollen season Contact with flowering birch Face and leg oedema, rhinitis, conjunctivitis		0	
	May	0.5–1.71	Plant pollen season		May-December 2 × 1.8 per day	
	June	0.5–0.8	Plant pollen season Fever 38 °C			
	September	0.0–1.0	Rhinitis, cough			
	November	0.8–1.0	Rhinopharingitis			
	December	0.47–7.5	Rhinopharingo-bronchitis, fever 40 °C			
2020 7 nephrological consultations	January	0.3–1.0	Fever 38 °C, rhinitis, cough	January–December various doses 25.0–5.0 per day	January–December 2 × 1.8 per day	
	February	0.3–1.0	Rhinitis, cough			
	April	1.4	Plant pollen Season—contact with flowering birch			
	June	0.3–1.0	Plant pollen season			
	September	0.3–1.14	Consumption of pancakes with white cheese			
	October	0.3–1.0	Consumption of cheese pizza, runny nose			
	December	2.06–4.49	Common cold, fever 40 °C			
Introduction of the allergist’s recommendations: dietetic and antiallergic treatment. Continuation of pharmacological INS treatment.						
2021 6 nephrological consultations	January	0.0–1.0	Abdominal complaints	January–December various doses 20.0–1.0 per day	January–December various doses 2 × 1.8 per day than systematic dose reduction to 1 × 0.5	Dietary treatment (Milk-free and hypoallergenic diet). Antiallergic treatment.
	March	0.2–1.0	Common cold, fever 40 °C			
	July	1.0–2.0	Plant pollen season. Dietetic error (blackcurrant juice) fever, vomiting			
	August	0.7–1.0	Herpes simplex infection			
	September /October	1.0–6.82	Dietetic error (consumptions of ice cream and a cookie containing milk)			
	December	0.15–1.0	COVID-19 infection, rhinosinusitis, cough			
Remission of INS clinical symptoms. End of INS pharmacological treatment						
2022 3 nephrological consultations	January	0.3	Varicella infection	0	0	
	May	0.3	Dietetic error (apple-carrot juice)	0	0	
	December	trace proteinuria	Consumption of broth containing beef	0	0	
2023	Long-term remission of INS clinical and laboratory symptoms. Continuation of dietary treatment. Antiallergic treatment as necessary.					

## Data Availability

Data sharing not applicable.

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
