# Peer review of "Allergy to Cow’s Milk Proteins and Other Allergens—An Unrecognized Co-Factor of Idiopathic Nephrotic Syndrome in Children or a Factor Interfering with the Treatment of This Disease? A Case Report"

_reports, 2023, doi:10.3390/reports6020025_

Round 1

Reviewer 1 Report

This is a case report of steroid dependent nephrotic syndrome receiving mycophenolate mofetil and long term steroids with associated multiple allergies mainly to cow milk and seasonal allergens. Authors have made an interesting association. I suggest they need to discuss the effect of Mycophenolate as a confounder in this patient because of its continued use here and its effects on T lymphocytes and previously described effect on atopic dermatitis and food allergies. Also since multiple allergens were involved keeping cows milk as the prime culprit here may not be warranted.

Author Response

Thank you for your review and comments.

Answer 1.

According to international and Polish guidelines on the principles of treatment of NS in children, Mycophenolate mofetil (MMF) is an immunosuppressive drug used in steroid-resistant INS (INS SR) or in INS with frequent recurrences. The drug is used to maintain INS remission when there are side effects of steroid therapy or when side effects occur after using mainly cyclosporine (CsA), as well as other medications used in the treatment of INS (e.g. tacrolimus - TAC, cyclophosphamide - CYC, others).

The immunosuppressive effect of MMF is associated with a reduction in the number of T and B lymphocyte populations in the patient, but it does not affect the synthesis of pro-allergic cytokines. In the described case, MMF was added to the therapy regimen in the 6th month of the INS SR treatment, during the third phase of the disease (May 2019), which occurred when trying to reduce prednisone from a dose of 30mg/24h to a dose of 30mg/48h.

Further treatment of the INS SR was then recommended: prednisone 30 mg/day with a gradual reduction of the dose by 5 mg every 6 days + MMF (Cell Cept) at a dose of 2x360 mg/24h = 2x 1.8 ml. Despite treatment modifications, by the end of 2019 the girl had 4 episodes of INS exacerbation and further hospitalizations. In 2020, relapses of proteinuria (without the need for hospitalization) occurred on average every 2 months. This type of combination treatment, which did not prevent recurrences of INS symptoms, was continued until December 2020 (1.5 years). To sum up: MMF added to steroid therapy did not have a beneficial effect on the course of the INS and its long-term use did not lead to stabilization of INS in the presented case.

Answer 2.

In the period preceding the onset of INS (2.5 years of age), the girl often had symptoms typical of an allergic disease (symptoms concerning the nose, conjunctiva, throat, and bronchi). For this reason, the girl was often given anti-allergic drugs. During the same period, her twin brother was treated for cow's milk allergy (CMA) = positive family history.

In 2021, the girl was confirmed to have specific antibodies against cow's milk proteins (asIgE class 2). This result was the basis for recognizing CMA as the primary cause of allergic symptoms. Persistent allergy to cow's milk proteins initiated a generalized allergic process in the child which also involved the kidneys.

The clinical confirmation of the relationship between allergy to cow's milk proteins and exacerbation of INS symptoms was the appearance of proteinuria after the child consumed broth cooked with beef (homology of allergens of cow's milk proteins and beef). Intensification of proteinuria after a walk among pollinating birch, as well as the appearance of proteinuria and edema after eating, among other foods, apples, bananas or carrots (despite the use of INS treatment), argued for the additional participation of a secondary sensitization to pollens of plants and trees (the so-called pollen-food syndrome - PFS).

Considering the probable relationship between the allergic process and INS exacerbations in the described patient (primary CMA and secondary pollen and food allergy), a diagnostic dairy-free and hypoallergenic elimination diet and anti-allergic treatment was recommended, during an allergological consultation. The girl has been in INS remission for 2.5 years, she is still under the supervision of a pediatric nephrologist, she does not receive any nephrological drugs and adheres to a balanced elimination diet.

Reviewer 2 Report

This publication discusses the relationship between idiopathic nephrotic syndrome (INS) and atopic-allergic processes, which can lead to the development of a systemic allergic inflammatory process affecting the kidneys. It presents a case of a child with INS who also had a cow's milk protein allergy and hypersensitivity to other allergens, and explores the effect of an elimination diet and anti-allergic medications on the course and treatment of INS. Overall, the publication highlights the importance of considering and addressing coexisting allergic processes in the management of INS.

Comments to the Author

It is important to note that a single case report or a small case series may not provide conclusive evidence and should be interpreted with caution. Larger studies with more patients are needed to validate the findings.

The presence of an allergic process in patients with idiopathic nephrotic syndrome is not a new concept and has been extensively studied in the past. Therefore, this study may not provide new insights into the relationship between allergies and nephrotic syndrome.

The use of an elimination diet and anti-allergic medications as a treatment for idiopathic nephrotic syndrome coexisting with an allergic process is not a standard treatment protocol. Therefore, the efficacy and safety of this treatment approach should be further evaluated in larger clinical trials before it can be recommended as a standard treatment option.

It is important to consider the limitations of the study, such as potential biases or confounding factors, to properly interpret the results.

Author Response

Thank you for your review and comments.

Answer 1 &2.

The importance of the diagnosis and appropriate treatment of the concomitant allergic process in patients with Idiopathic Nephrotic Syndrome Steroid Resistant (INS SR) was first noticed by Sandberg and Genova (ref. 36, 37). These authors observed, in some patients, frequent relapses and exacerbations of INS despite the use of INS treatment according to accepted guidelines. Only the diagnosis of CMA coexisting with INS in these patients and the introduction of a dairy-free diet resulted in temporary remission of Idiopathic Nephrotic Syndrome Steroid Sensitive (INS SS) (Sandberg) or remission of INS SR (Genova). 

In a Polish study from 1992, Sieniawska (ref. 10) presents a course of INS in 17 children. Coexistence of allergy to cow's milk proteins was confirmed in 6 of the subjects and the use of a dairy-free elimination diet for a year changed the course of their INS. In 2 patients there was a complete remission of INS, which was associated with the acquisition of tolerance to cow's milk proteins. Unfortunately, these studies were not continued. The author agrees with the opinion of the reviewer that the described case is a supplement to previously described observations.

However, one gets the impression that this research topic has not become the subject of further in-depth analysis. Another case described by the author serves as a reminder that the coincidence of INS and CMA is possible and may adversely affect the treatment and course of INS in children. Proper diagnosis and treatment of food allergy or other allergic disease coexisting with INS may favourably modify the further course of INS in some children, as evidenced by the case described by the author.

Answer 3.

The author of the publication did not violate or question the principles of standard treatment protocols for children with INS. The elimination diet recommended in the described case and the supportive antiallergic treatment were used in parallel to INS treatment recommended by a nephrology team. Since the introduction of the elimination diet and antiallergic treatment, the doses of steroidal drugs were systematically reduced until the complete discontinuation of steroids as well as MMF in accordance with the recommendations of the lead nephrologist treating the child. The patient has been in full INS remission for 2.5 years.

Answer 4.

The author is aware of the limitations of this study. The use of dietary and anti-allergic treatment of an allergic disease coexisting with INS is not intended by the author of the publication as a recommendation for standard treatment of this disease, but a suggestion to take into account the possibility of the coincidence of both diseases, which may adversely affect the course and treatment of INS in some children. Due to the observed failures in the treatment of INS in children and the undesirable side effects of the medications used, each such case requires, in the author's opinion, an individual diagnostic and therapeutic approach. This may offer a chance for a more optimal treatment of nephrotic syndrome in some children.

Reviewer 3 Report

This is a case report about an old topic of the association of Allergy and NS. Although author tried to illustrate the association, the mechanism was still not clear. If the author want to make the conclusion more explicit, a schematic figure is needed. Overall, the article is redundant, there are many issues to be improved.

1 As a  case report , CARE checklist should be followed.

2 The changes of the indicators and treatments should be shown by a line chart.

3 There are no results of IL13 IL-4, IL-5 of the case.

4 The discussion of differences of IGE and non IGE induced allergy on NS was not clear.

Author Response

Thank you for your review and comments.

Answer 1. The publication was prepared according to the guidelines posted on the journal's website: REPORTS - Case Report General Consideration

Answer 2. See the Table 1 which may be included in the manuscript if the editors agree.

Table 1. The course of INS in the presented patient before the allergological consultation and after the introduction of the allergist's recommendations (elimination diet, anti-allergic treatment)

Answer 3.  Measurements of interleukins IL-4, IL-13, Il-5 are not used in routine diagnostic procedures in patients with suspected allergic disease. In the described case, the observations were made during the patient's standard antiallergic treatment. The basis for the diagnosis of IgE-dependent allergy to CMA and pollen and food allergy coexisting with INS were the results of allergological tests: elevated total serum IgE values (tIgE 380.0IU/ml, then tIgE-140.0 U/ml) and a positive allergy test (asIgE test against cow's milk proteins, hen's egg, and dog's epidermis). Diagnosis of pollen and food allergy (pollen-food syndrome – PFS) was based on the history: symptoms occurring during the pollen and flowering season of plants (runny nose, lacrimation, and cough) associated with the consumption of e.g. apples, bananas, carrots, which were the basis for other specialists to suspect early childhood asthma symptoms in the child.

Answer 4.

The criteria for the diagnosis of IgE-dependent allergy are: elevated or high levels of IgE in the patient's serum, the presence of allergen-specific antibodies (asIgE) against the tested allergens (food, airborne) or a positive result of Skin Prick Tests with specific allergens. Positive results only indicate sensitization of the patient's body, in which immunoglobulin E plays a pathogenic role.  It should be remembered that the sensitization itself is an asymptomatic process, but without the primary sensitization of the body, the development of an allergic disease will not occur. With repeated exposure to the same allergen, a certain group of sensitized people will develop an allergic disease. This happens when immunologically competent cells of the subpopulation of T lymphocytes are activated and then differentiated into effector cells, e.g. Th2. Th2 cells stimulated by the sensitizing allergen produce pro-allergic cytokines (IL-4, IL-13, IL-5) involved in the synthesis of IgE and the development of systemic IgE-dependent allergic inflammation (allergic inflammation type 1). These cytokines, acting on selected organs and systems of the human body, lead to their functional disorder, which is manifested by specific allergic symptoms.

In non-IgE-allergy, the level of serum IgE is very low or within physiological limits. The presence of asIgE against the tested allergens is also not detected, the results of skin prick tests are also negative.  IgE-mediated allergic inflammation, similarly to IgE-mediated reactions, is caused by contact of the human body with environmental allergens. However, it takes place at the subcellular level in specific organs (type 2 allergic inflammation). In the development of this inflammation are involved, among others, ILC2 lymphoid cells. This cells, when stimulated by an allergen, produce pro-inflammatory cytokines. These are mainly cytokines IL-25 and IL-33 capable of damaging the epithelial cells of the mucous membranes of certain organs and skin, but the cytokines IL-13 and IL-5 are also produced in this process. The role of IL-13 in IgE allergy and non-IgE-mediated allergy and nephrotic syndrome was discussed in the introduction to this publication. In an active allergic process, excessive production of IL-13 leads to the permanent expression of the CD80 costimulatory molecule by podocytes. This particle is involved in the presentation of the allergen to immunologically competent cells in the human body.  Its expression on podocytes disturbs both their anatomical and functional structure (podocyte effacement). This leads to impaired filtration of plasma proteins in the glomeruli and the onset of clinical and laboratory symptoms of INS.

According to available literature, the presence of IL-13 as a mediator of an allergic process results in the expression of the CD80 molecule on podocytes in patients with INS and a concomitant IgE-dependent or IgE-independent allergy. The course of this process is presented in Figure 2 in the cited publication (Ref. 4. Maher Abdel-Hafez, et al. Idiopathic Nephrotic Syndrome and atopy; is there a common link? Am J Kidney Dis. 2009;54(5):945-953).

In the author's opinion, in order to end/limit the expression of the CD80 molecule by podocytes in patients with INS and a concomitant allergic process, this [allergic] process should be recognized/diagnosed and controlled/managed.

Round 2

Reviewer 3 Report

I still have some minor comments.

1 Long-term remission  of INS symptoms should be specified. What was the extent of proteinuria?

2 The changes of the indicators  should be reported as numbers in table 1.

3 What is a hypoallergenic elimination diet (elimination of certain fruits, vegetables, nuts, artificially colored sweets and drinks)?

4 The mechanism of this hypoallergenic elimination diet as non-IgE-mediated allergic reactions to some other non-dairy foods should be specified.

5 What does Mykofenolan mofetilu mean in table 1?

Author Response

1.Long-term remission of INS symptoms should be specified. What was the extent of proteinuria?

From the moment the girl was diagnosed with an allergic process coexisting with INS (December 2020) and the introduction of an elimination diet and anti-allergic treatment, a noticeable remission of INS symptoms began: systematic decrease in the frequency of recurrence of INS clinical symptoms, i.e. edema, and laboratory symptoms, i.e. proteinuria. The maximum level of proteinuria in April 2019 was 14.5 g/l. In December 2020, before the allergological consultation, the level of proteinuria ranged between 2.06 - 4.49 g/l.  After commencing treatment, the proteinuria decreased steadily until it completely disappeared. In 2021 – 2022, episodes of proteinuria occurred sporadically - during infections or after dietary mistakes (maximally to 1g/l). Proteinuria values are presented in Table 1.

From January 2021, the nephrologist reduced the combined doses of INS treatment, i.e. prednisone and Mycophenolate mofetil (MMF), until the complete discontinuation of these drugs in December 2021. Steady remission of INS symptoms is observed as of 2023.

2.The changes of the indicators should be reported as numbers in the Table 1.

Changes in proteinuria are presented in Table 1.

  1. What is a hypoallergenic elimination diet (elimination of certain fruits, vegetables, nuts, artificially, colored sweets and drinks)?

A hypoallergenic diet is a diagnostic and therapeutic procedure consisting of a temporary elimination of certain allergenic foods from the diet of a patient who presents specific allergic symptoms. It is especially applicable in patients with non-IgE-mediated allergy. Temporary elimination of potentially harmful foods and then their reintroduction to the patient's diet allows confirming or excluding the relationship of an allergic reaction with the consumption of a specific food. The most common allergenic foods in children are: milk, egg, wheat, nuts, fruits, vegetables, and dyes contained in sweets and drinks.

The interview regarding the course of INS in the presented case showed that the occurrence of INS exacerbations and episodes of proteinuria in the girl in 2019-2021 were associated by her mother with the child's consumption of milk foods, and certain fruits and vegetables. For this reason, the child was advised to temporarily eliminate certain potentially allergenic foods from the diet, i.e. a hypoallergenic diet.

4.The mechanism of this hypoallergenic elimination diet as non-IgE mediated allergic reactions to some other non-dairy foods should be specified.

The hypoallergenic diet also eliminates certain non-dairy products that may cause an allergic reaction in the patient. The argument for the recommendation to eliminate certain fruits and vegetables in the girl was the association of INS exacerbations and the occurrence of proteinuria with the consumption of these foods, especially during the pollen and flowering seasons.

In an IgE-independent reaction, synthesis of mediators of the allergic reaction, such as: IL-4, Il-13, Il-5, takes place in the body of the sensitized person at a subcellular level, which was described earlier in the publication. A hypoallergenic diet (including the elimination of certain fruits and vegetables) leads to inhibition of the synthesis of interleukin 13 and the expression of the CD80 molecule by podocytes.

  1. What does Mykofenolan meofetilu mean in Table 1.

Table content has been corrected to MMF (Mycophenolate mofetil). 

Round 3

Reviewer 3 Report

Good job.